

# On the impact of service-oriented patterns on software evolvability: a controlled experiment and metric-based analysis

Justus Bogner[1,2], Stefan Wagner[2] and Alfred Zimmermann[1]

[1] Herman Hollerith Center, University of Applied Sciences Reutlingen, Boeblingen, Baden-Wuerttemberg, Germany
[2] Institute of Software Technology/Software Engineering Group, University of Stuttgart, Stuttgart, Baden-Wuerttemberg, Germany

## ABSTRACT

**Background:** Design patterns are supposed to improve various quality attributes of software systems. However, there is controversial quantitative evidence of this impact. Especially for younger paradigms such as service- and Microservice-based systems, there is a lack of empirical studies.

**Objective:** In this study, we focused on the effect of four service-based patterns—namely *Process Abstraction*, *Service Façade*, *Decomposed Capability*, and *Event-Driven Messaging*—on the evolvability of a system from the viewpoint of inexperienced developers.

**Method:** We conducted a controlled experiment with Bachelor students ($N = 69$). Two functionally equivalent versions of a service-based web shop—one with patterns (*treatment group*), one without (*control group*)—had to be changed and extended in three tasks. We measured evolvability by the effectiveness and efficiency of the participants in these tasks. Additionally, we compared both system versions with nine structural maintainability metrics for size, granularity, complexity, cohesion, and coupling.

**Results:** Both experiment groups were able to complete a similar number of tasks within the allowed 90 min. Median effectiveness was 1/3. Mean efficiency was 12% higher in the treatment group, but this difference was not statistically significant. Only for the third task, we found statistical support for accepting the alternative hypothesis that the pattern version led to higher efficiency. In the metric analysis, the pattern version had worse measurements for size and granularity while simultaneously having slightly better values for coupling metrics. Complexity and cohesion were not impacted.

**Interpretation:** For the experiment, our analysis suggests that the difference in efficiency is stronger with more experienced participants and increased from task to task. With respect to the metrics, the patterns introduce additional volume in the system, but also seem to decrease coupling in some areas.

**Conclusions:** Overall, there was no clear evidence for a decisive positive effect of using service-based patterns, neither for the student experiment nor for the metric analysis. This effect might only be visible in an experiment setting with higher initial effort to understand the system or with more experienced developers.

Corresponding author
Justus Bogner, justus.bogner@iste.uni-stuttgart.de

# INTRODUCTION

One important concern for enterprise software in today's digital and fast-moving world is the ability to quickly adapt to new or changing functional or cross-functional requirements. This concern is addressed by the software quality attribute *evolvability* (sometimes also referred to as modifiability or changeability): the degree of effectiveness and efficiency with which a software system can be modified to adapt or extend it (*Rowe, Leaney & Lowe, 1998*; *International Organization For Standardization, 2011*). Several benefits related to this quality attribute were achieved with the rise of service-oriented computing (SOC) (*Papazoglou, 2003*), such as loose coupling, isolation of the service implementation behind business-relevant interfaces, or convenient reuse and composition. While Service-Oriented Architecture (SOA) (*Erl, 2005*) is still an important architectural style, Microservices (*Newman, 2015*; *Fowler, 2015*) gain more and more popularity as a flexible, lightweight, and decentralized service-oriented variant.

One frequently used instrument to enhance modifiability is the application of *design patterns*. Employing these established solution blueprints for recurring problems is especially common with object-oriented systems. There is, however, also a significant amount of patterns specifically designed for service-based and even Microservice-based systems (*Erl, 2009*; *Rotem-Gal-Oz, 2012*; *Richardson, 2018*). One issue with design patterns is that their relationship with quality attributes (QAs) is often complex and governed by trade-offs. Moreover, while the benefits of patterns for QAs like modifiability seem plausible in theoretical and qualitative studies (*Bogner, Wagner & Zimmermann, 2019*), quantitative empirical evidence for their effectiveness is of a more controversial nature. In scientific literature, we find studies that do report a positive impact on QAs, studies that do not, and studies that do so under certain conditions or only for selected patterns (*Garzás, García & Piattini, 2009*; *Hegedűs et al., 2012*; *Ali & Elish, 2013*). Awareness of and familiarity with the concrete patterns is often discussed as a prerequisite for their effectiveness.

Since most of these studies are concerned with object-oriented or architectural patterns and there is very little empirical research on service-oriented patterns and modifiability, we conducted a controlled experiment to partially address this gap. A total of 69 students in two groups changed and extended two functionally equivalent versions of a service-based web shop system (one pattern version, one non-pattern version) while the time was measured for each task. Independent of this experiment, we also collected structural maintainability metrics (e.g. size, coupling, cohesion) for both system versions to have a foundation for a second comparison. The **research objective** for this study can therefore be summarized in the following way:

**Analyze selected service-oriented patterns**
**For the purpose of improving modifiability**
**With respect to effectiveness, efficiency, and structural metrics**
**From the viewpoint of inexperienced software developers (students)**
**In the context of a service-based web shop system**

We follow the basic structure of the reporting guidelines for experiments in software engineering as proposed by *Jedlitschka, Ciolkowski & Pfahl (2008)*. The remainder of the paper first presents the technical background ("Background") by elaborating the concept of (service-oriented) design patterns and discussing related work in the area. After that, we describe the experiment design ("Experiment Design") and present the experiment results ("Experiment Results") followed by the metric results ("Metric Analysis"). In the sections thereafter, we provide possible interpretations ("Discussion") and discuss limitations ("Threats to Validity"). Lastly, we compile the most important lessons learned ("Lessons Learned from the Experiment") and conclude with a summary as well as potential future work ("Conclusion").

## BACKGROUND

To understand the motivation behind this study, two topics need to be explained in greater detail: namely patterns as an instrument of software design as well as their relation to QAs, for which we present related work in the area.

### Design patterns

The idea of design patterns originated from the construction and city building language of *Alexander, Ishikawa & Silverstein (1977)*, who conceptualized a network of solution blueprints. The concept was adapted to several other domains including computer science and is very popular in software engineering and software architecture. As such, a pattern is a proven and established solution to a recurring design problem that is documented in a technology-agnostic form and can be implemented in many similar yet not completely identical ways. The documentation is often systematic and standardized within a pattern language and includes for example attributes like context, problem, forces, solution, or related patterns. While the most famous examples are the object-oriented "Gang of Four" design patterns of *Gamma et al. (1994)*, there are meanwhile patterns for software architecture (*Buschmann et al., 1996*), enterprise applications (*Fowler, 2002*), message-based integration (*Hohpe & Woolf, 2003*), or cloud computing (*Fehling et al., 2014*).

There is also a significant body of patterns in the field of SOC. Most of these have been conceptualized for the context of SOA. Prominent examples are the patterns by *Erl (2009)*, *Erl et al. (2012)*, *Rotem-Gal-Oz (2012)*, or *Daigneau (2011)*. They are usually on an architectural level and are for example, concerned with service inventories, communication, and composition, but can also be focused on the design of an individual service. Even though a significant number of SOA patterns seems to be also applicable to Microservices (*Bogner, Zimmermann & Wagner, 2018*), the first pattern languages for the younger service-based architectural style are emerging (*Richardson, 2018*). Furthermore, several of these Microservices patterns have existing ancestor patterns from SOA or other contexts.

### Related work

One primary driver for the use of patterns is their impact on QAs like availability, performance, or modifiability. Several studies have been conducted to analyze this complex and controversial relationship.

In the context of object-oriented design, *Garzás, García & Piattini (2009)* conducted a controlled experiment with 47 students who had to analyze and modify four UML class diagrams. One group worked on a standard and straightforward design model while the other had a semantically equivalent version that contained design rules (e.g., "Dependencies between classes must be implemented with abstractions.") and patterns (e.g. *State* or *Composite*). Understandability (via questions) and modifiability (via extension tasks) were measured. Results showed that the latter version with rules and patterns was more difficult to understand (58% less time and 15% more correct answers for the non-pattern version). For modifiability, no statistically significant difference in efficiency could be identified.

*Hegedūs et al. (2012)* used a probabilistic quality model based on ISO/IEC 9126 to analyze the maintainability of more than 300 revisions of the Java GUI framework JHotDraw, which employs well-known object-oriented patterns. Every usage of design patterns in JHotDraw is documented with JavaDoc and there are a lot of revisions that only introduce patterns. The authors conclude from the analysis that the introduction of additional patterns increased the overall maintainability in the system. They measured a strong correlation ($r$-value: 0.89) between pattern-line-density and maintainability.

A broader view on the impact of the "Gang of Four" design patterns on software quality is given by *Ali & Elish (2013)*. Their comparative literature analysis of 17 empirical studies revealed that only four QAs and only a small subset of the patterns have been examined. Moreover, no general consensus concerning the impact could be reached (positive, neutral, or negative). Interestingly, for maintainability, evolution, and change-proneness, the overall tendencies concerning the impact of the analyzed patterns were negative.

In the domain of architectural patterns, *Kassab, El-Boussaidi & Mili (2012)* analyzed the impact of the patterns *Pipes and Filters*, *Layers*, *Model View Controller*, and *Broker* on the two QAs performance and security. They determined the quantitative effect of patterns on QAs via the proxy of architectural tactics. From these results, they concluded for example, that *Model View Controller* is best suited for performance while being least suited for security and that the *Layers* pattern is most accommodating for security.

*Riaz, Breaux & Williams (2015)* conducted a systematic mapping study with the goal to characterize the research design of empirical studies with human subjects about the application of software patterns. Maintenance was the most frequent context with 16 of 30 studies. Nearly half of the studies were concerned with object-oriented design patterns (14). Efficiency and correctness were the most common measures for evaluating the pattern application. The authors also report that differences in experiment design make it difficult to compare the results and that several studies fail to mention limitations as well as how they minimized the threat of biases.

In the context of service orientation, *Galster & Avgeriou (2012)* performed a theoretical qualitative mapping of ~80 service-based design patterns to the QAs of the S-Cube Quality Reference Model via force resolution maps (impact from −2 to +2). They reported that 53 QAs from the very detailed S-Cube model were not addressed by the patterns. Most mapped QAs were performance and scalability. Since S-Cube does not include some

important QAs, they also used ISO/IEC 9126. For maintainability, they identified a total of 12 patterns.

Lastly, *Palma et al. (2014)* analyzed the impact of service-based patterns and anti-patterns on maintenance and evolution by collecting historical software development meta data (*# of Changes* and *Code Churn*) for the FraSCAti system. They observed that services involved in patterns required less maintenance effort. However, this effect was not statistically significant. Services with anti-patterns on the other hand were found to need significantly more maintenance effort, especially for instances of *God Component* or *Service Chain*.

The presented related work gives an overview of the complex relationship between patterns and QAs and the controversial evidence. Not many empirical quantitative studies exist for service-based patterns in general and their modifiability in particular, which is why we aim to bring additional quantitative insights into this relationship with the results of the first controlled experiment as well as a metric-based analysis.

## EXPERIMENT DESIGN

The **research goal** for our experiment was to analyze if selected service-based patterns have a significant impact on the evolvability of a system in terms of the completion of modifications within a given time (effectiveness) and the time needed per modification (efficiency). The **experiment object** was a simple service-based web shop system that has been specifically constructed for this experiment[1]. It consists of several RESTful Java services for example, customers, orders, products, and notifications and a web based frontend. Data persistence and unnecessary Create Read Update Delete (CRUD) operations have not been fully implemented. As such, the system is reasonably close to a real world web shop, but is still of manageable complexity for an experiment. The online shop domain was chosen because most people are somewhat familiar with it from personal experience. Moreover, it is very common to implement such systems in a service-oriented way.

We created two functionally equivalent versions of this web shop. One version was built in an "ordinary" way (see Figs. 1 and 2) while the other version was designed with several service-based patterns that are believed to be beneficial for modifiability, for example, *Process Abstraction* and *Service Façade* (see Figs. 3 and 4). Table 1 lists the selected patterns together with their source, intended effect, and relevant task number. In general, the pattern version of the system exhibits a higher base complexity (e.g., more services, special patterns), but has been intentionally prepared for the nature of the task modifications through the used patterns. We chose these patterns because their theoretical benefit for evolvability is well documented.

While professional software developers who are familiar with service-based systems and patterns could be fitting experiment subjects, we instead opted for more inexperienced developers, that is, students. First, it is very difficult to convince a large number of software professionals to spend two hours of their valuable time for free on seemingly meaningless coding. And second, if the patterns' advantages materialize even with inexperienced developers that have little or no conscious knowledge of them, their effect on evolvability must be substantial. However, while it is common to use students in software engineering

[1] See https://github.com/xJREB/research-modifiability-pattern-experiment for source code, task descriptions, survey questions, data set, and analysis results. Zenodo mirror for non-source code artifacts: DOI 10.5281/zenodo.3340971.

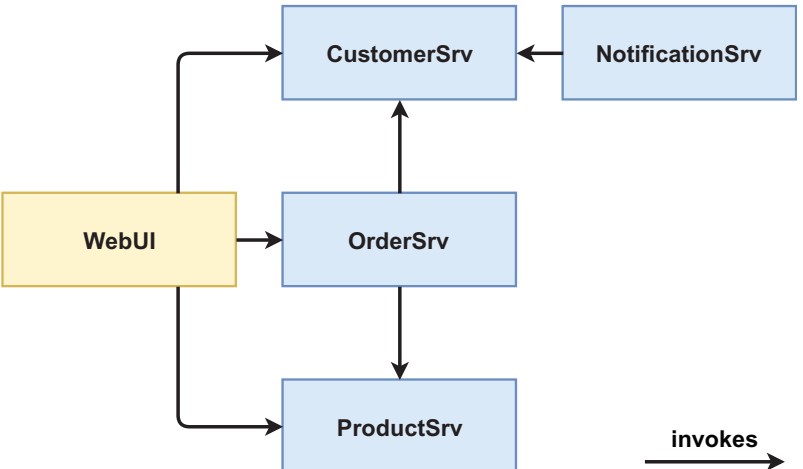

**Figure 1** Version #1 pre-experiment: initial architecture of non-pattern version.

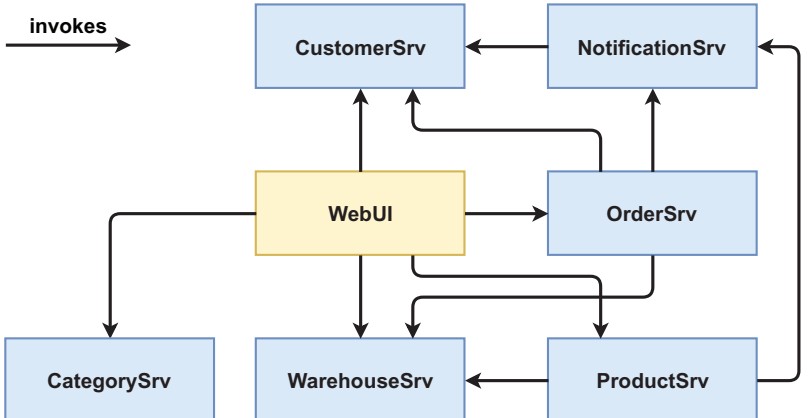

**Figure 2** Version #1 post-experiment: final architecture of non-pattern version.

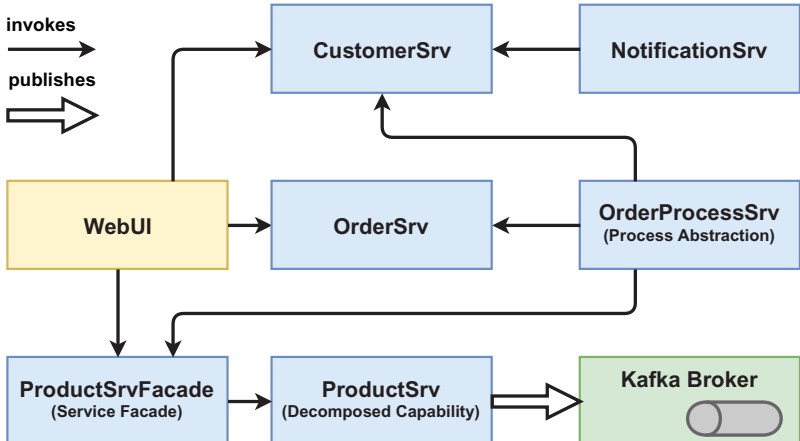

**Figure 3** Version #2 pre-experiment: initial architecture of pattern version.

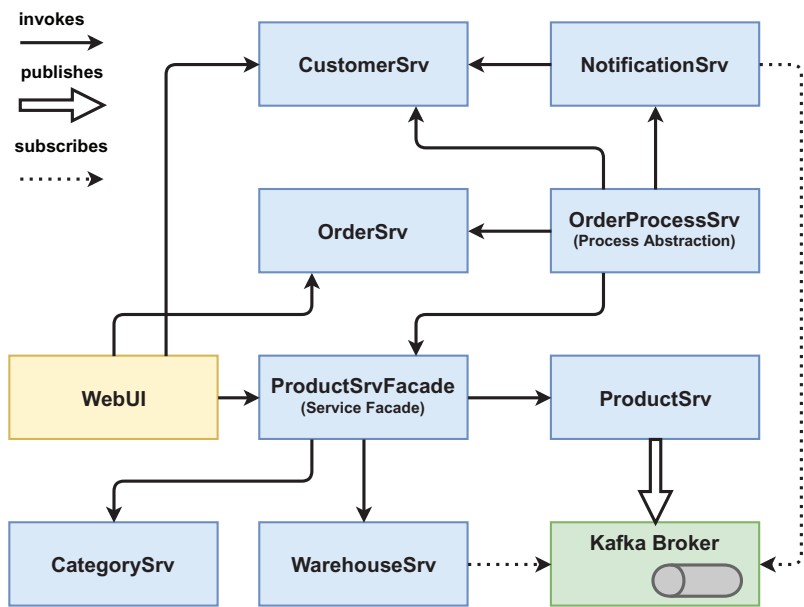

**Figure 4 Version #2 post-experiment: final architecture of pattern version.**

**Table 1 List of applied patterns in system version #2.**

| Pattern name | Source | Intended effect | Task |
|---|---|---|---|
| Process abstraction | Erl (2009) | Details of the order process are abstracted in composition controller; changes can be made in central location (OrderProcessSrv) | #1 |
| Service façade | Erl (2009) | Shields the ProductSrv against all consumers; changes to the interface only have to be addressed at the façade | #2 |
| Decomposed capability | Erl (2009) | Large and incohesive ProductSrv is prepared for future decomposition by internally isolating domain concepts | #2 |
| Event-driven messaging/ inversion of communications | Erl (2009) and Rotem-Gal-Oz (2012) | ProductSrv publishes events instead of directly calling other services; decoupling of producers and consumers | #3 |

experiments, one must be more careful when generalizing the results to the population of all developers. On the plus side, students are often fairly homogeneous participants. Finally, several studies have shown that using students instead of software professionals may affect the results only marginally in a lot of cases (*Host, Regnell & Wohlin, 2000*; *Salman, Misirli & Juristo, 2015*; *Falessi et al., 2018*).

Our **experiment subjects** therefore were Bachelor students ($N = 69$) that needed to participate in an experiment as part of the "Introduction to Software Engineering" lecture (mostly 2nd and 3rd semesters). Students could choose one of two experiments based on a short description. Data collection was anonymous and experiment performance had no influence on the students' grades. Participating in the experiment without data collection was also possible to pass the course. During experiment execution, students assigned

| Task | Task description |
|------|-----------------|
| **Table 2** | **List of experiment tasks.** |
| #1 | Adjust the web shop ordering process (customer credit rating check, minimum available number of products) and extend it with an additional process step (sending of an email via the `NotificationSrv`). |
| #2 | Decompose the large `ProductSrv` into three smaller services: a `ProductSrv` to manage only product domain entities, a `CategorySrv` to manage product categories, and a `WarehouseSrv` for product availability. |
| #3 | Implement a new process triggered in response to adding a new product to the database. The new process sends out emails and adds the new product to a marketing database. |

themselves randomly and unknowingly to one of the two groups by choosing a seat in the PC pool room: either the non-pattern version #1, that is, the *control group*, or the pattern version #2, that is, the *treatment group*. As **experiment materials**, students were provided with a fully configured virtual machine in the university PC pool room. They had documentation and task descriptions in both digital and printed form. They were allowed to use internet search in case of issues. A web interface with automatic tests for validating the completion of each of the three tasks was provided as well. Students were advised to measure their own time per task with a stopwatch of their choosing.

Participants had to solve a total of **three tasks** that depended on each other (see Table 2). In the first task, the ordering process of the web shop system should be adjusted (e.g., customer credit rating check) and extended with an additional process step (sending of an email via the `NotificationSrv`). Version #2 had been prepared with the *Process Abstraction* pattern so that all changes had to be implemented in the `OrderProcessSrv` as opposed to in three different services like in version #1. In the second task, the large `ProductSrv` had to be decomposed into three smaller services: a `ProductSrv` to manage only product domain entities, a `CategorySrv` to manage product categories, and a `WarehouseSrv` for product availability. Version #2 incorporated the *Decomposed Capability* pattern to ease the decomposition as well as the *Service Façade* pattern that shielded the former `ProductSrv` from consumers. In the final task, a new process in response to adding a new product to the database should be implemented (sending out email and adding the new product to a marketing database). Version #2 provided message-based communication via an Apache Kafka broker that allowed publishing a `NewProductEvent`, which implements the patterns *Event-Driven Messaging* and *Inversion of Communications*. Please refer to the folders `workspace-version1/_exercises` or `workspace-version2/_exercises` in our GitHub repository (https://github.com/xJREB/research-modifiability-pattern-experiment) for the complete task descriptions.

As **response variables** for our experiment, we analyzed effectiveness and efficiency (duration per task). Effectiveness of a participant was measured as the percentage of the three tasks he/she successfully completed within the 90 min, that is, 0%, 33%, 67%, or 100%. Efficiency was recorded in seconds or as not available if the task was not completed. Median effectiveness per group was only calculated and tested for the total of all three tasks. For mean efficiency, we additionally also analyzed and compared every individual task to derive the effect per pattern. While these two response variables also depend on the skill of the participants, they can characterize the systems' evolvability if the two groups

**Table 3** Pairs of Experiment Hypotheses (each in the form of an alternative hypothesis $H_{1ij}$ and its null hypotheses $H_{0ij}$, where $i$ represents the research goal identifier and $j$ the counter for more than one hypothesis per goal; effectiveness: $i = 1$, efficiency: $i = 2$); $E_{vx}$ denotes the effectiveness for version $x$; $D_{vx}$ denotes the task durations for version $x$; $D_{vx,ty}$ denotes the task durations for version $x$ and task $y$.

| | Alternative hypothesis | Null hypothesis |
|---|---|---|
| Effectiveness | $H_{111}$: More tasks for the pattern version #2 of the system can be completed within the given time than for the non-pattern version #1: $\text{median}(E_{v2}) > \text{median}(E_{v1})$ | $H_{011}$: There is no difference in how many tasks can be completed for both versions of the system: $\text{median}(E_{v1}) \approx \text{median}(E_{v2})$ |
| Efficiency (task duration) | $H_{121}$: It takes less time to complete task#1 for the pattern version #2 of the system than for the non-pattern version #1: $\text{mean}(D_{v2,t1}) < \text{mean}(D_{v1,t1})$ | $H_{021}$: There is no difference in the time it takes to complete task#1 for both versions of the system: $\text{mean}(D_{v1,t1}) \approx \text{mean}(D_{v2,t1})$ |
| | $H_{122}$: It takes less time to complete task#2 for the pattern version #2 of the system than for the non-pattern version #1: $\text{mean}(D_{v2,t2}) < \text{mean}(D_{v1,t2})$ | $H_{022}$: There is no difference in the time it takes to complete task#2 for both versions of the system: $\text{mean}(D_{v1,t2}) \approx \text{mean}(D_{v2,t2})$ |
| | $H_{123}$: It takes less time to complete task#3 for the pattern version #2 of the system than for the non-pattern version #1: $\text{mean}(D_{v2,t3}) < \text{mean}(D_{v1,t3})$ | $H_{023}$: There is no difference in the time it takes to complete task#3 for both versions of the system: $\text{mean}(D_{v1,t3}) \approx \text{mean}(D_{v2,t3})$ |
| | $H_{124}$: It takes less time to complete a task for the pattern version #2 of the system than for the non-pattern version #1: $\text{mean}(D_{v2}) < \text{mean}(D_{v1})$ | $H_{024}$: There is no difference in the time it takes to complete a task for both versions of the system: $\text{mean}(D_{v1}) \approx \text{mean}(D_{v2})$ |

are large enough and roughly equal in skill. The **predictor variable** was the group or system version (i.e., control or treatment group). It was either #1 for the non-pattern version or #2 for the pattern version.

To formalize our research objective, we constructed five **experiment hypotheses** $H_{1ij}$ and their respective null hypotheses $H_{0ij}$, where $i$ denotes the research goal identifier and $j$ the counter if there is more than one hypothesis per goal (see Table 3). For effectiveness ($i = 1$), we have one hypothesis ($j = 1$) while for efficiency ($i = 2$), we have four ($1 \leq j \leq 4$, $j \in \mathbb{N}$), namely one per individual task and one for the complete set of tasks at once. Since we have five hypotheses, this also means that we need Bonferroni correction for the significance level of our hypothesis tests to account for the increased probability of type I errors. The necessary significance level $\alpha$ therefore is calculated by dividing the desired significance level by the number of hypotheses, that is, $\alpha = 0.05/5 = 0.01$.

The **experiment execution** took place in the following way. To prepare participants for the experiment, an introductory presentation was given on a separate day (45 min). A total of 55 of the 69 students attended (~80%). In this session, the structure and procedure of the experiment were explained. We also described the data collection and analysis process. Furthermore, an introduction to the basic concepts of SOC and RESTful HTTP services was given to ensure a common base level of knowledge. Lastly, the details of the experiment workspace were presented, e.g. Ubuntu VM, Eclipse Integrated Development Environment (IDE), directory structure, build scripts, task validation.

The actual experiment took place over the course of the week following the introductory presentation in slots of 10–20 students. In such a slot (~2 h), there was first a short

introduction (5 min) explaining the procedure and agenda. Details of what would be measured and what data would be collected via the post-experiment survey were presented. We pointed out that experiment performance had absolutely no influence on the grade. Data collection would be anonymous and participant IDs could not be linked to student IDs. Following these explanations, we asked if there were any questions regarding this process and if it was acceptable for everybody (verbal consent). We also specifically pointed out the option to participate in the experiment without data collection.

After that, there was a familiarization period (15 min) during which students should get comfortable with the workspace and the system by reading documentation and playing around with the IDE and the build scripts. This period was followed by the actual task execution with time measurement. Participants had 90 min to complete all three tasks. A web-based evaluation application with automated tests was provided to check for successful task completion. Participants recorded their own time with a stopwatch and paused upon successful validation of a task via the evaluation UI. An experiment administrator was then notified to verify the completion and to document the duration. The timer was then reset and the next task began. After solving all three tasks or after 90 min, participants finally filled out a short web-based survey with questions about the perceived difficulty per task, personal information (e.g. course of study and semester), and their self-reported experience with for example Java, service-based systems, and patterns. Their participant ID and system version was also recorded to relate it to the task durations. It was not possible to identify the student by name via the participant ID, which guaranteed the anonymity of the results. Please refer to the repository for the full list of questions. After completing this survey, participants finished the experiment slot and were allowed to leave.

## EXPERIMENT RESULTS

For the analysis, the documented task duration measurements per participant were first combined with the exported survey results via the participant ID. We then divided the resulting data set into the two groups (version #1 and version #2) and analyzed it with **descriptive statistics**. Initially, we wanted to ensure that both versions had comparable characteristics and experience, which is the case in most areas (see Table 4). On average, group #1 with 36 participants and group #2 with 33 participants were of roughly the same study program distribution and semester (~2.5). When comparing programming experience and self-reported skill, group #2 seems to have been slightly more experienced. More participants of group #1, however, attended the introductory presentation (~13% points more), a factor that was correlated with effectiveness (Kendall's tau: 0.346, *p*-value: 0.0019). The standard deviation for most attributes was also similar in both groups and fairly low in general (e.g. around or below 3.0 for most 10-point ordinal scale questions). Therefore, the set of participants could be considered as sufficiently homogeneous. So all in all, the two groups were similar enough to assume equal conditions for an effectiveness and efficiency comparison with respect to the treatment, that is, the patterns.

With 1/3, median **effectiveness** was identical for both groups. Overall, 48 of 69 participants (~70%) were able to solve task #1, a total of 26 of these additionally solved task

Table 4 Group characteristics and self-reported experience (SD represents the standard deviation; for experience questions based on the 10-point ordinal scale, 1 represents "not experienced" while 10 represents "very experienced").

|  |  | Group #1 (no patterns) | Group #2 (patterns) |
|---|---|---|---|
| Participants |  | 36 (52%) | 33 (48%) |
| B.Sc. Business information systems |  | 9 (25%) | 7 (21%) |
| B.Sc. Computer science |  | 4 (11%) | 4 (12%) |
| B.Sc. Software engineering |  | 22 (61%) | 20 (61%) |
| Other study programs |  | 1 (0.03%) | 2 (0.06%) |
| Introduction attendance |  | 31 (86%) | 24 (73%) |
| Semesters | Mean | 2.36 | 2.45 |
|  | SD | 1.15 | 1.09 |
| Years of programming experience | Mean | 2.13 | 2.70 |
|  | SD | 1.80 | 2.63 |
| Java experience (1–10) | Mean | 6.03 | 6.55 |
|  | SD | 2.36 | 2.18 |
| Web development experience (1–10) | Mean | 3.61 | 4.30 |
|  | SD | 2.86 | 3.19 |
| Service-based systems experience (1–10) | Mean | 1.58 | 2.42 |
|  | SD | 1.30 | 2.68 |
| Design patterns experience (1–10) | Mean | 3.94 | 4.61 |
|  | SD | 2.93 | 3.16 |
| Service-based patterns experience (1–10) | Mean | 1.86 | 2.85 |
|  | SD | 1.55 | 2.48 |
| All experience-related attributes (1–10) | Mean | 3.41 | 4.15 |

#2 (~38%), and only 17 participants finished all three tasks (~25%). Roughly 30% were not able to solve any task, namely 10 out of 36 for group #1 (27.8%) and 11 out of 33 for group #2 (33.3%). The self-reported difficulty/complexity per task (1–10) was also fairly similar for both groups. The only notable exception for this was task #3 which was perceived as 2.61 points less difficult by the pattern group #2 (6.15 vs 3.54 points). When filtering only for the 17 participants who actually finished this task, the difference is nearly identical (2.64 points), even though the estimated difficulty is lower (4.86 vs 2.22).

When analyzing participant **efficiency**, that is, duration for task completion, we observed that the mean duration per completed task for the total of all three tasks was about 12% lower for the pattern group #2 (00:32:45 vs 00:28:50). The analysis per individual task revealed that this is caused by task #2 and #3: group #2 needed on average ~22% less time for task #2 and ~51% for task #3 respectively. Task #1, on the other hand, took group #2 ~15% more time to complete. Table 5 lists the detailed results for this. The efficiency difference can also be conveniently observed in a boxplot that shows the statistical distribution for task duration (in seconds) grouped by system version and task number (see Fig. 5).

The next step was **hypothesis testing**, that is, analyzing if the differences between the groups are statistically significant so that we can reject the null hypotheses. To prepare

**Table 5 Result measures per group (SD represents the standard deviation; for difficulty questions based on the 10-point ordinal scale, 1 represents "not difficult" while 10 represents "very difficult").**

| | | Group #1 (no patterns) | Group #2 (patterns) |
|---|---|---|---|
| Participants that solved task#1 | | 26 (72%) | 22 (67%) |
| Participants that solved task#2 | | 15 (42%) | 11 (33%) |
| Participants that solved task#3 | | 7 (19%) | 10 (30%) |
| Effectiveness | Median | 1/3 | 1/3 |
| | 1st Quartile | 0/3 | 0/3 |
| | 3rd Quartile | 2/3 | 3/3 |
| Reported difficulty for task#1 (1–10) | Mean | 3.52 | 3.35 |
| | SD | 2.26 | 1.92 |
| Reported difficulty for task#2 (1–10) | Mean | 5.60 | 5.43 |
| | SD | 2.63 | 2.91 |
| Reported difficulty for task#3 (1–10) | Mean | 6.15 | 3.54 |
| | SD | 2.60 | 2.82 |
| Duration per individual task | Mean | 00:32:45 (1,965 s) | 00:28:50 (1,730 s) |
| | SD | 00:17:09 (1,029 s) | 00:19:11 (1,151 s) |
| Duration for task#1 | Mean | 00:30:32 (1,832 s) | 00:35:10 (2,110 s) |
| | SD | 00:18:41 (1,121 s) | 00:22:03 (1,323 s) |
| Duration for task#2 | Mean | 00:39:41 (2,381 s) | 00:30:47 (1,847 s) |
| | SD | 00:15:38 (938 s) | 00:12:27 (747 s) |
| Duration for task#3 | Mean | 00:26:07 (1,567 s) | 00:12:45 (765 s) |
| | SD | 00:09:23 (563 s) | 00:04:32 (272 s) |
| Duration for All three tasks (in total) | Mean | 01:15:30 (4,530 s) | 01:03:39 (3,819 s) |
| | SD | 00:10:20 (620 s) | 00:18:18 (1,098 s) |

the selection of a suitable statistical test, we first used the Shapiro–Wilk test to check if our samples were non-normally distributed. For all samples, the $p$-value was substantially smaller than 0.05, so we had to reject the null hypothesis that our data came from a normal distribution. We therefore needed a non-parametric test that could handle non-normal distributions. The Mann–Whitney $U$ test (also known as Wilcoxon–Mann–Whitney test) fulfills this requirement. It checks the null hypothesis that the probability is equal that a random value from one group is less than or greater than a random value from another group. We used an exact implementation correcting for ties that were likely to happen for the effectiveness test (only four different values: 0/3; 1/3; 2/3; 3/3).

Since median effectiveness of both groups is identical (1/3), the resulting $p$-value for the hypothesis test is much too large (0.5903). This means we cannot reject $H_{011}$ and therefore have no support for $H_{111}$ that more exercises can be completed for pattern version #2. For efficiency, we first tested all three tasks at once ($H_{124}$) where we identified a mean difference of about 12%. The resulting $p$-value of 0.0496 is barely below the 0.05 level, but since we need a significance level of 0.01 due to multiple hypotheses, this is still too large. We therefore cannot confidently reject our null hypothesis $H_{024}$, that is, we cannot support $H_{124}$ that the pattern group #2 was overall more efficient on a statistically

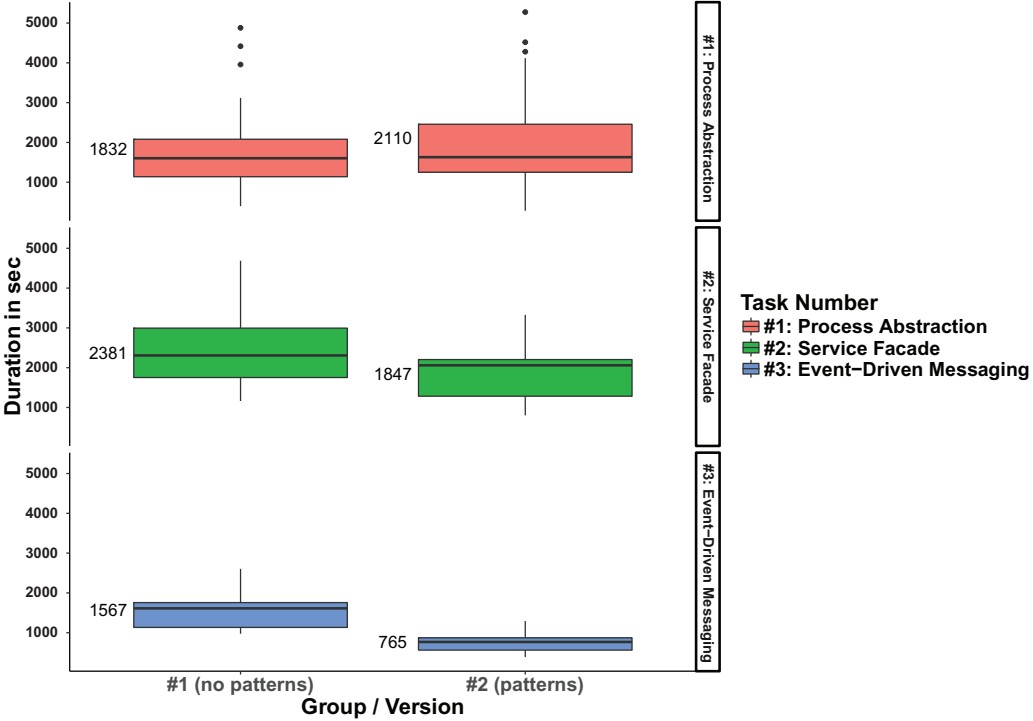

**Figure 5 Boxplot comparison of the duration per version and task.**

significant level. When performing the same test for the three tasks individually, the resulting $p$-values are 0.741 (task#1), 0.082 (task#2), and 0.0006 (task#3) respectively. This means that with our Bonferroni-corrected significance level of $p \leq 0.01$ (desired significance level divided by the number of hypotheses $\Rightarrow \alpha = 0.05/5 = 0.01$) we could only reject $H_{023}$ and identify support for $H_{123}$ (task#3, patterns: *Event-Driven Messaging/ Inversion of Communications*). A post-hoc power analysis for our only successful hypothesis test (i.e. the probability that the test correctly rejected the null hypothesis) revealed that the statistical power is sufficient (0.931). As pointed out, all other four null hypotheses ($H_{011}$, $H_{021}$, $H_{022}$, $H_{024}$) could not be rejected.

## METRIC ANALYSIS

For a second comparison of the two system versions, we chose and collected measurements for nine different maintainability metrics (see Table 6) related to structural design properties such as size, complexity, and coupling from the final systems (post-experiment optimal solutions, see Figs. 2 and 4). Some of these metrics are pretty simple (e.g. *# of Services* or *Lines of code* (*LOC*)). Since a number of more sophisticated maintainability metrics specifically designed for service orientation have been suggested in scientific literature (*Bogner, Wagner & Zimmermann, 2017*), we also chose some of these (e.g. *service interface data cohesion* (*SIDC*) or *absolute dependence of the service* (*ADS*)). All metrics (except for *# of Services*) were collected per service and then aggregated to the system level with aggregation functions such as *SUM, MEAN,* or *MAX*. Before we

**Table 6 List of the nine selected maintainability metrics to analyze both versions.**

| Metric name | Design property | Source |
| --- | --- | --- |
| # of services | Size | – |
| Lines of code (LOC) | Size | – |
| Weighted service interface count (WSIC) | Size | *Hirzalla, Cleland-Huang & Arsanjani (2009)* |
| LOC/WSIC | Granularity | – |
| Cyclomatic complexity (CC) | Complexity | *McCabe (1976)* |
| Service interface data cohesion (SIDC) | Cohesion | *Perepletchikov (2009)* |
| Absolute dependence of the service (ADS) | Coupling | *Rud, Schmietendorf & Dumke (2006)* |
| Relaxed ADS | Coupling | – |
| Absolute importance of the service (AIS) | Coupling | *Rud, Schmietendorf & Dumke (2006)* |

describe the detailed metric results, we briefly present the selected metrics, explain our rationale for choosing them, and point out how they were collected.

## Metric definitions

In the area of **size and granularity**, we selected four metrics. The premise with these metrics is that a smaller system with appropriate service granularity (not too many services, not too large services) will be easier to understand and maintain. The first one was *# of Services*, which already is a proxy for system size and therefore does not need to be aggregated. So, if we assume unchanged granularity, fewer services are generally easier to grasp and maintain. We manually derived *# of Services* from the final architecture diagrams. In both versions, the WebUI was counted as a service while in the pattern version #2, the Kafka broker was not counted as a service, as it does not contain custom code and is an infrastructure component.

As a second metric, we selected the prevalent *LOC* metric that we collected for each service via the static analyzer SonarQube (https://www.sonarqube.org). We then created system-level *LOC* aggregates with *SUM*, *MEAN*, *MEDIAN*, and *MAX*. Since *LOC* is sometimes seen as controversial (e.g. if several programming languages are involved), we also selected a volume metric specifically designed for service orientation, namely the *weighted service interface count* (*WSIC*) by *Hirzalla, Cleland-Huang & Arsanjani (2009)*. *WSIC* represents the count of publicly available operations in a service interface with possible weights for different types of operations (e.g. synchronous and asynchronous). We used the standard weight of 1, which is basically the same as *# of Operations*. Values for *WSIC* were automatically derived from the existing OpenAPI (https://github.com/OAI/OpenAPISpecification) files with a self-developed analysis tool. Like *LOC*, *WSIC* was also aggregated with *SUM*, *MEAN*, *MEDIAN*, and *MAX*.

To gain further granularity-related insights in addition to the means and medians of our two volume metrics, we also calculated their ratio, that is, *LOC/WSIC*. For a given service, this represents the number of LOCs that are on average necessary to provide a single

operation. Larger values for *LOC/WSIC* mean a service has large and potentially complex operations. This metric was aggregated with *MEAN*, *MEDIAN*, and *MAX*.

As a measure for **complexity**, we selected *cyclomatic complexity* (*CC*) from *McCabe (1976)* as a traditional source code metric. While some suggestions for service-based complexity metrics like *response for operation* (*Perepletchikov et al., 2007*) or *Message Entropy* (*Mateos et al., 2017*) exist, tool support for their collection is not available and they are hard to calculate manually. Despite its criticisms (*Ostberg & Wagner, 2014*), we therefore relied on the widely used *CC* metric that was gathered for each service via SonarQube. We then subsequently aggregated the values with *MEAN*, *MEDIAN*, and *MAX*. Lower values for *CC* suggest that a service is easier to understand and maintain.

In the context of **cohesion**, we chose the *SIDC* metric proposed by *Perepletchikov (2009)*. *SIDC* rates the cohesion of a service interface in percent based on the input and response parameter data types of its operations. If a service interface operates on the same or only a small set of data abstractions (e.g., CRUD operations for a `Customer` entity), the values for *SIDC* will be closer to 100% and the service will be easier to analyze and change. We used the already mentioned OpenAPI analysis tool to automatically calculate the percentage values for *SIDC*. These values were then aggregated with *MEAN*, *MEDIAN*, and *MIN* (as lower values are worse).

The last maintainability-related design property that we wanted to measure was **coupling**. In the service-oriented space, "loose coupling" is a prevalent theme aiming to reduce the number and strength of service dependencies and therefore preventing or mitigating ripple effects on changes (*Pautasso & Wilde, 2009*). We therefore chose three metrics to analyze the degree of coupling in both versions, where lower values mean less coupling and therefore increased maintainability. All coupling metrics were manually derived from the final architecture diagrams that also include the dependencies between services. Moreover, the same aggregations were used for all of them, namely *MEAN*, *MEDIAN*, and *MAX*.

First, we selected the *ADS* metric proposed by *Rud, Schmietendorf & Dumke (2006)*. For a given service, *ADS* represents the number of other services that this service depends on, that is, of which it invokes at least one operation. For pattern version #2, dependencies to the Kafka broker were counted as well. Since *ADS* treats every static dependency exactly in the same way, we also collected values for an adjusted variant of *ADS* that respects looser types of coupling. In that sense, *Relaxed ADS* works like *ADS*, except that dependencies to a *Service Façade* or the Kafka broker were counted with 0.5 instead of 1. The rationale for this was that these two patterns are introduced as intermediaries to reduce coupling. As a consequence, dependencies to them should not be weighted in the same way as direct invocations of services. In version #1 of the system, the values for *Relaxed ADS* are therefore exactly the same as for *ADS*. Only in the pattern version #2, the values for the two variants differ.

The third and last coupling metric, also proposed by *Rud, Schmietendorf & Dumke (2006)*, is complementary to *ADS*, namely the metric *absolute importance of the service* (*AIS*). For a given service, *AIS* represents the number of other services that have a dependency to this service, that is, that invoke at least one of its operations. Since the invocation origin is not really important, we did not gather a *Relaxed AIS* variant.

**Table 7 System-level metric results per version (Change in % refers to the change of the metric value from V1 to V2; a positive percentage means V1 is better, a negative one means V2 is better; in cases with a change of at least 15%, the better version is marked with a colored background).**

| Metric | Aggregate | Version #1 (no patterns) | Version #2 (patterns) | Change in % |
|---|---|---|---|---|
| # of services | – | 7 | 9 | 28.6 |
| LOC | SUM | 2,555 | 3,303 | 29.3 |
| | MEAN | 365 | 367 | 0.5 |
| | MEDIAN | 324 | 362 | 11.7 |
| | MAX | 591 | 668 | 13.0 |
| WSIC | SUM | 32 | 47 | 46.9 |
| | MEAN | 5.33 | 5.88 | 10.2 |
| | MEDIAN | 5 | 5 | 0.0 |
| | MAX | 9 | 12 | 33.3 |
| LOC/WSIC | MEAN | 79.09 | 105.77 | 33.7 |
| | MEDIAN | 77.33 | 71.73 | −7.2 |
| | MAX | 112.40 | 373.00 | 231.9 |
| CC | MEAN | 48.57 | 48.44 | −0.3 |
| | MEDIAN | 50.00 | 48.00 | −4.0 |
| | MAX | 93 | 100 | 7.5 |
| SIDC | MEAN | 0.370 | 0.374 | −1.2 |
| | MEDIAN | 0.40 | 0.40 | 0.0 |
| | MIN | 0.16 | 0.16 | 0.0 |
| ADS | MEAN | 1.57 | 1.56 | −1.0 |
| | MEDIAN | 1 | 1 | 0.0 |
| | MAX | 5 | 4 | −20.0 |
| Relaxed ADS | MEAN | 1.57 | 1.28 | −18.7 |
| | MEDIAN | 1 | 0.5 | −50.0 |
| | MAX | 5 | 3.5 | −30.0 |
| AIS | MEAN | 1.57 | 1.22 | −22.2 |
| | MEDIAN | 1 | 1 | 0.0 |
| | MAX | 3 | 3 | 0.0 |

## Metric results

To compare the two system versions, we only present the aggregated system-level metrics in this paper (see Table 7). For a detailed comparison of the underlying service-level metrics, please refer to the metric evaluation spreadsheet in our repository (https://github.com/xJREB/researchmodifiability-pattern-experiment/blob/master/_results/metric-analysis.xlsx) that includes the measurements for each service.

When comparing the system-level metrics for size, we immediately see that pattern version #2 is larger. It has two more services (`OrderProcessSrv` and

`ProductSrvFaçade`) and therefore consists of ~29% more LOC and ~47% more operations. Even though the Kafka broker is also new in version #2, it is not counted as a service. While *MEAN* and *MEDIAN* for the size metrics are only slightly worse or stay roughly the same in version #2, the *MAX* value for *WSIC* increases by ~33% (from nine to 12 operations). This is due to the newly introduced `ProductSrvFaçade` that relays calls to the `ProductSrv`, `CategorySrv`, and `WarehouseSrv`. Lastly, the introduction of the `OrderProcessSrv` in version #2 impacted the *LOC/WSIC* ratio. While the *MEDIAN* is slightly better for version #2 (~7%), both the *MEAN* value (~34%) and the *MAX* value (~232%) are worse. The reason for this is that the `OrderProcessSrv` provides only a single operation while simultaneously consisting of slightly above average LOC.

For both complexity and cohesion, our chosen metrics do not show much differences between the two versions. *CC* aggregates are very similar, with the only notable difference being a slightly larger *MAX* value (~8%) for version #2. This is caused by adding the messaging functionality to the `NotificationSrv`. Aggregates for *SIDC* are even more similar, which suggests that the patterns at hand do not influence service cohesion all that much.

The only design property that seems slightly better in pattern version #2 is coupling. While the *MEAN* and *MEDIAN* aggregates of *ADS* stay the same in both version, the *MAX* value in version #2 has been reduced by 20%: the `ProductSrvFaçade` shields the services behind it so that the `WebUI` has one dependency less (four instead of five). If we treat looser forms of coupling differently, version #2 improves even further. For *Relaxed ADS*, all aggregates are better in the pattern version (*MEAN* by ~19%, *MEDIAN* by 50%, and *MAX* by 30%), because the Kafka broker and `ProductSrvFaçade` reduce the weight of service dependencies. Finally, even though the *MEDIAN* and *MAX* aggregates for *AIS* are the same in both versions, the *MEAN* value is improved by ~22% in version #2. This is caused by the *Event-Driven Messaging/Inversion of Communications* patterns. The Kafka broker does not actively call services, but services have to publish or subscribe to it. Therefore, the *SUM* values for *ADS* and *AIS* would also be different in version #2, even though they would be the same in version #1.

## DISCUSSION

From the experiment results, we could not derive support for the majority of our hypotheses that service-based patterns had a positive impact on participant effectiveness and efficiency. The mean difference in duration was only significant for task #3 in isolation. We offer three main interpretations to explain these results. One straightforward possibility is that the patterns of task #1 (*Process Abstraction*) and task #2 (*Service Façade* and *Decomposed Capability*) were simply not effective enough to enhance the modifiability of the system under test. Their theoretical benefit for the chosen evolution scenario did not (or only partially) translate to a practical setting. Only *Event-Driven Messaging/Inversion of Communications* from task #3 led to a significant advantage for the system's evolvability. While this seems plausible at first sight and our chosen patterns may certainly differ in their effectiveness, we believe that our second and third interpretations are more likely.

Another explanation for the results may be that the effect of familiarization and experience was stronger for the pattern version #2. As they progressed through the tasks,

participants naturally grew more accustomed to the system and workspace environment. Since the pattern version #2 exhibited a higher base complexity (more services, more inter-service communication, more build scripts to start before task validation), participants of this group were initially slowed down more than the control group. Over the course of the experiment, they gradually adjusted to this so that the effectiveness of the chosen patterns for the evolution scenario could unfold (slightly in task#2 and fully in task#3). This effect could have been weakened by randomizing the order of tasks per participant. Unfortunately, this was not possible because tasks depended on each other.

We also offer a possible third explanation in conjunction with the familiarization effect. The patterns' effect on modifiability seems to have been influenced by whether participants had conscious knowledge of and experience with patterns beforehand. When we analyzed existing correlations between effectiveness and self-reported experience-related factors, we observed that both knowledge of general design patterns as well as service-oriented patterns was more strongly correlated with effectiveness in the pattern group #2 than in group #1: about 19% more for general patterns (Kendall's tau: 0.485 vs 0.579) and about 242% more for service-oriented ones ($r$-values: 0.093 vs 0.318). Years of programming experience, for example, was similarly correlated with effectiveness in both groups ($r$-values: 0.509 vs 0.497). So using students instead of experienced professionals who have worked with patterns before seems to have hurt treatment group #2 more. The potential modifiability-related benefit of a pattern may be lessened or even negated by its complexity and impact on understandability, if the participant is not familiar with the pattern. Potential support for this can be found by narrowing down the sample for both groups to only the best participants. When we filter for only the 26 students that solved at least task #1 and task #2 (effectiveness $\geq$ 67%), the mean efficiency difference increases: instead of ~12%, participants of pattern group #2 now needed ~31% less time per task.

Overall, the results suggest that the theoretical evolvability-related advantage of service-oriented patterns is difficult to replicate in controlled experiments: familiarity with the system and experience with the selected patterns seem to have an impact on the patterns' effectiveness. For inexperienced developers unfamiliar with the system, the additional complexity introduced by the patterns seems to reduce or even outweigh the theoretical positive effect on modifiability. Implications of such a conclusion could be that appropriate documentation of used service-oriented patterns as well as thorough pattern-focused initial training of new developers become all the more important to ensure a long-term and sustainable effect of patterns on software evolvability.

With respect to the metric analysis, we observed that the pattern version #2 is worse in the area of size and granularity and better for coupling. Our chosen complexity and cohesion metrics are not impacted by the patterns. When counting only changes in metric values of at least 15%, version #1 is superior for six size and granularity aggregates (*# of Services*, $LOC_{SUM}$, $WSIC_{SUM}$, $WSIC_{MAX}$, $LOC/WSIC_{MEAN}$, and $LOC/WSIC_{MAX}$), while version #2 is better for five coupling aggregates ($ADS_{MAX}$, *Relaxed* $ADS_{MEAN}$, *Relaxed* $ADS_{MEDIAN}$, *Relaxed* $ADS_{MAX}$, and $AIS_{MEAN}$). However, three of these five improvement areas for version #2 are aggregates of *Relaxed ADS*, a metric that we specifically

constructed for the loose coupling nature of the used patterns. Without *Relaxed ADS*, the pattern version would only be significantly better for two coupling aggregates ($ADS_{MAX}$ and $AIS_{MEAN}$).

All in all, the comparison of structural metrics showed no decisive maintainability improvements in the pattern version, which seems to be in line with the experiment results. The increased system size and slightly worsened granularity in version #2 may support our interpretation that it took longer in this group until the familiarization effect kicked in. More services and operations meant that participants were potentially under higher cognitive load and required more effort to get familiar with the system. Lastly, the partially improved coupling measurements in the pattern version could explain why participants in this group required less time for task#2 and especially task#3: these tasks relied on the patterns *Service Façade* and *Event-Driven Messaging*, which are both related to decoupling.

## THREATS TO VALIDITY

Threats to validity have to be examined in several areas of this empirical study. With respect to **construct validity**, our operationalized experiment measure (namely the time necessary to implement a feature) seems valid to represent the construct in a practical real-world setting. Efficiency is one of the most used measures for software engineering experiments and, in contrast to maintainability metrics, it is not a structural approximation of this quality attribute. Effectiveness, that is, the degree to which participants solved all tasks within the given time frame, is a similarly popular measure in software development, even though efficiency is more relevant in a real-world industry setting. Most often, it is not the question, *if* something can be implemented, but *how long* it will take. Lastly, the results of the metric analysis rely on the maintainability prediction quality of the chosen metrics. Several of these metrics (e.g. *LOC* or *CC*) are well-known and have been extensively studied, but especially some of the younger service-oriented metrics have not been evaluated in or accepted by large-scale industry environments. So, while the chosen design properties seem very fitting for a service-oriented context, the metrics selected to represent them may be of differing quality. Similarly, only a limited number of metrics was analyzed and there is always the possibility for more or different metrics.

**Internal validity** is concerned with how much the treatment was actually responsible for the observed effect and if there were unknown factors that influenced the results. During the experiment discussion, we already mentioned the observed impact of the participants' pattern knowledge on the effective modifiability of the patterns. A possible solution to this could have been to only accept participants with a base-level of pattern experience or to include a small lecture on service-oriented patterns in the introductory presentation. We also already described the familiarization effect for later tasks, which makes it harder to analyze the effectiveness of each individual pattern. Task randomization as a solution to this was not possible because the task order was important in the constructed evolution scenarios.

Furthermore, participants were forced to use the provided experiment workspace via a virtual machine. While most students should be somewhat familiar with Eclipse, their

preferred IDE, OS, or build tool may be different from the provided environment. This could have hindered their development speed. We have to assume that this effect was similar for each group and task. Moreover, participants were allowed to ask questions, if they were not related to the actual coding, but to the experiment environment (e.g., IDE, build tool, and evaluation UI).

A last potential threat in this area is the general coding ability of participants that may tamper with the results: students that are very slow/fast in general work with similar speed regardless of the group. Since participants only worked on one version in our experiment, an uneven distribution of very slow/fast students could have affected the groups' mean efficiency. While our population of 69 has a smaller risk to be influenced by this than for example, 20 participants and our post-experiment survey did not reveal major experience differences between the groups, the self-reported nature of the comparison still leaves some room for issues. Possible solutions could have been to conduct a pilot evaluation study with the participants to divide them into two groups of similar skill or to let participants work on tasks from both groups in a rotating manner. Both solutions were not used because of time and effort constraints.

Concerning the metric analysis, we relied on the correctness of the collected measurements. More complex metrics were gathered automatically with tool support while simple metrics were manually derived (e.g. from architecture diagrams) and double-checked across researchers. Even though it is not very likely, there still remains the small possibility of incorrect metric values that may have clouded the analysis.

**External validity** refers to the generalizability of the results to the target population and setting. While the usage of students may be a valid proxy for the desired target population in many cases, our experiment was very challenging for Bachelor students. Only ~25% solved all three tasks and ~30% could not solve any task. We also hypothesize that the missing degree of pattern experience influenced the treatment group's effectiveness and efficiency. Therefore, we expect that a replication with experienced software professionals from industry would yield different results. However, such a replication with a sufficient number of participants is extremely difficult to organize.

We created both versions of the web shop system as close to the real world as possible. Nonetheless, controlled experiment tasks are still inherently of a somewhat artificial nature with the potential for subjective bias. The experiment results are also bound to the used programming language (Java) and service-based communication style (RESTful HTTP). Moreover, we designed the tasks with the intuitive feeling that the pattern version #2 might be faster to change, because the patterns are perfectly suited for the changes at hand. The benefit of a pattern will always heavily depend on the specifics of the evolution scenario that is performed. In conjunction with this, developers are usually decently familiar with the system that they extend. So, in a real world software maintenance scenario, the benefits of modifiability mechanisms and patterns often manifest over a long period of time with increasing developer familiarity with the system.

The artificial construction of the two system versions may also have impacted the reliability of the metric-based analysis. After all, we evaluated the internal quality of artifacts that were created by ourselves, which leaves possibilities for researcher bias.

To mitigate these threats, metric collection was performed by external research assistants (see "Acknowledgments") and the final set of metrics was not known during the system construction period. Nonetheless, using several existing industry or open source systems for the metric-based analysis of patterns would have provided more objective results. In the case of our study, however, the goal of the evaluation with metrics was to provide a second perspective on the two experiment system versions.

Lastly, one must be very careful to generalize from the results of four patterns, a total of 69 student participants, and nine metrics to all patterns and software developers. The controlled experiment presents first empirical insights for the modifiability of selected service-oriented patterns with inexperienced human participants, while the metric study provides additional structural insights that aim to further the understanding of the patterns' effects. However, many more similar studies should follow to either support or reject the conclusions of this research.

## LESSONS LEARNED FROM THE EXPERIMENT

We experienced a number of limitations with our experiment design that hindered our means for analysis and interpretation. To aid future controlled experiments in the area of design patterns' impact on modifiability and to prevent researchers from repeating the same mistakes, we documented some lessons learned. First, tasks should not depend on each other. This allows to **randomize task order per participant** to lessen the familiarization effect and analyze the impact of individual patterns. Furthermore, you can then set fixed maximum durations per task which ensures participants work on all tasks. This may obviously decrease the overall number of solved tasks though, especially if they are difficult.

Another suggestion is to conduct a pilot experiment with similar tasks to rate participants. This rating can then be used to randomly draft individuals to **ensure similarly skilled groups**. As a less time-consuming alternative, a survey with self-reported skill can be used. If a pre-experiment study is not possible, tasks could be designed to allow participants to work on both versions of the system in alternating fashion. An even number of tasks should be chosen in this case.

Lastly, it is strongly advised to **ensure participants' familiarity with the patterns**. Otherwise their effect will be reduced. In combination with this, the most realistic software maintenance/evolution scenario requires that **participants are already familiar with the system** to a certain degree. This could be achieved by using an existing system and its developers. A second version would need to be constructed though. If no fitting existing system is identified and time allows it, a long-term familiarization period with artificial systems could be used before the actual experiment.

## CONCLUSION

To analyze the impact of service-oriented design patterns on software evolvability, we conducted a controlled experiment with 69 Bachelor students. Participants had to change and extend a service-based web shop system in two functionally equivalent versions over the course of three tasks. We measured effectiveness and efficiency per group.

While median effectiveness was the same for both groups (1/3), we saw differences for the mean efficiency, that is, mean duration per task. Participants in the treatment group with patterns were about 12% faster (00:32:45 vs 00:28:50), but due to Bonferroni correction not at a statistically significant level ($p$-value: 0.0496). When analyzing each individual task, we found only the group difference for task #3 (pattern: *Event-Driven Messaging*) to be of a significant nature ($p$-value: 0.0006). Here, participants in the treatment group needed about 51% less time.

During the subsequent analysis of the two system versions with nine maintainability metrics, the pattern version #2 exhibited worse measurements in the area of size and granularity and better measurements for coupling, even though the most improved coupling metric was specifically designed for the patterns' type of dependency (*Relaxed ADS*). Complexity as well as cohesion measurements were similar between the two versions. Overall, we did not observe decisive maintainability metric improvements in the pattern version, which seems to be in line with the experiment results.

Our interpretation of these results is that we have no clear indication that three of the four selected service-based patterns were beneficial for evolvability. We theorize, however, that the additional volume introduced by the patterns hindered participants to leverage their modifiability-related benefits at first, which seems to be supported by the size and granularity metrics. Over the course of the experiment, participants became more and more familiar with the system and the patterns, which allowed the treatment group to get a slight edge in task #2 and finally produced full statistical significance in task #3. The implications of these results are that documentation and training of used service-based patterns should not be neglected in software maintenance and evolution scenarios.

With respect to possible future work, we already mentioned the lack of empirical quantitative research on service-oriented patterns and QAs (in our case evolvability). It is therefore necessary that future research provides additional support in this area. Many patterns for SOA and also some for Microservices are available and one study can only cover so many of them. Moreover, additional research could also aim to confirm and quantify the impact of developers' pattern experience on the effectiveness of the patterns. Additionally, the metric-based analysis of patterns could be extended to existing industry or open source systems to mitigate the construction bias. As an alternative, several practitioners or external researchers could implement systems with these patterns to allow for a more objective analysis. To support such endeavors and to enable potential replication studies, we shared all artifacts related to the experiment and metric analysis on GitHub (https://github.com/xJREB/researchmodifiability-pattern-experiment) and Zenodo (DOI 10.5281/zenodo.3340971) (source code, task descriptions, survey questions, result data sets, analysis script).

## ACKNOWLEDGEMENTS

We kindly thank Daniel Graziotin from the University of Stuttgart as well as Maximilian Jager from the University of Mannheim for the fruitful discussions about our paper and specifically about the used statistical methods. Furthermore, we thank Aretina Iazzolino, Philipp Meyer, and Daniel Quack (all from the University of Stuttgart)

for their diligent support with the metric analysis. Lastly, we are very grateful for the constructive and detailed feedback provided by our reviewers.

### Funding

This work was funded by the Ministry of Science of Baden-Württemberg, Germany, for the doctoral programme "Services Computing." The funders had no role in study design, data collection and analysis, decision to publish, or preparation of the manuscript.

### Grant Disclosures

The following grant information was disclosed by the authors:
Ministry of Science of Baden-Württemberg, Germany.

### Competing Interests

Justus Bogner is not only a PhD student, but also a software engineer at DXC Technology. Stefan Wagner and Alfred Zimmermann have no potential competing interests.

### Author Contributions

- Justus Bogner conceived and designed the experiments, performed the experiments, analyzed the data, contributed reagents/materials/analysis tools, prepared figures and/or tables, performed the computation work, authored or reviewed drafts of the paper, approved the final draft.
- Stefan Wagner conceived and designed the experiments, analyzed the data, contributed reagents/materials/analysis tools, authored or reviewed drafts of the paper, approved the final draft.
- Alfred Zimmermann conceived and designed the experiments, authored or reviewed drafts of the paper, approved the final draft.

### Data Availability

Data is available at Zenodo:

Bogner, Justus. (2019). Data and Analysis Artifacts for Service-Based Evolvability Patterns (Experiment and Metrics) [Data set]. Zenodo. http://doi.org/10.5281/zenodo.3340972.

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
