# Peer review of "On the impact of service-oriented patterns on software evolvability: a controlled experiment and metric-based analysis"

_PeerJ Computer Science, doi:10.7717/peerj-cs.213_

## Round 0.1 · original submission · Major Revisions

As you can see, both reviewers found the paper interesting and relevant. However, Reviewer 2 has raised some important concerns regarding the study design as well as the analysis.

# Study Design #
The reviewer has some ethical concerns regarding the collection of personal (student) data. I largely concur with the reviewer here, but I think that fixing these problems is mostly a question of better documentation in the manuscript (please refer to the comments of Reviewer 2 for more details).

# Analysis #
Reviewer 2 also raises some concerns regarding the choice of statistical tests used in the study. Thankfully, the authors have provided the study data (which I appreciate a great deal!), which allowed Reviewer 2 to explore the data on their own. They found that the used data is not normally distributed, which makes the choice of hypothesis tests in the paper not appropriate. Re-analysing using a non-parametric test may be the best way forward here.

(the reviewer has also provided the analysis that they did, in the hope that this may be useful for your revision: https://figshare.com/s/7964b17bc270e549d448)

Note that I do not agree with the reviewer that hypothesis tests in general are unsuitable for this kind of work. However, I concur that the concrete choice of methods may not be appropriate given the nature of the data.

Reviewer 1 ·

Basic reporting

Within this submission, the authors present the results of a study they conducted. The study had the goal to show how service patterns influence the effectiveness and efficiency of software developers when working on service-oriented software.

While the results of this study may look disappointing at first sight, since most assumptions of the authors cannot be "proven" through the study, such results should also be published in order to give fellow researchers insights.

The paper is extremely well-written and easy to follow. I do about 50 reviews per year in different fields of distributed systems (from top journals to mediocre workshops), and this one is clearly within the top 5% of well-prepared submissions. I really wish that more authors would put a comparable effort into preparing their submissions.

Experimental design

As far as I can judge, the study is very well-conducted, following accepted best practices, and is conducted with the right amount of rigor. I would have expected that some aspects are taken into account in addition, e.g., if the sex or the age of the participants influence the outcomes of the experiments. It would be good if the authors clarify if such aspects are simply not mentioned because there was no significant impact of such parameters.

I especially like the self-critical approach of the authors and the discussion of the validity of the results and of the lessons learned from the study and experiments.

I would have expected that the authors do not only give (and discuss) average values in Tables 3ff, but also the standard deviations, since this gives very important information about the heterogeneity of the participants.

Validity of the findings

The most critical aspect of the conducted experiments is that the participants are very junior developers. In fact, I would even call Master students quite inexperienced developers, and even more so students which are mostly in their second or third Bachelor semester. Taking into account the rate of computer science students who stop their studies in Germany before getting their degree, it is very likely that a lot of the study participants have not the minimal level of knowledge needed to participate in such a study.

The goal mentioned in line 172ff (that it would be extremely meaningful if a positive effect could be shown even with inexperienced developers) is (with hindsight) therefore quite unrealistic. I completely understand that it is very difficult to get experienced developers to participate in such a study, but nevertheless, this significantly decreases the potential impact of this paper.

At the end of the day, it could be argued that the authors also "only" provide further insights for one particular group of developers, with an unclear outcome, i.e., similar to the studies mentioned in Lines 56ff. However, as can be seen from my overall recommendation, I still think the results should be accepted for publication in PeerJ CS.

Additional comments

* Line 50: I wouldn't necessary use "battle-tested" in order to describe that design patterns are well-accepted in software engineering. That's not necessarily scientific language.
* Please make use of vector graphics in order to improve readability and scalability of the figures.
* Section 4: I am not sure, if it's always the percentage that is meant here by the authors. In some cases, it might also be the percentage points instead. For instance, in line 248, is "(~13% more)" correct or should it rather be "~13 percentage points more"?
* The authors should carefully check their literature list, which is very heterogeneous with regard to layout and content. There is a lot of information missing, or double information. The authors should make sure that every single literature list entry provides the same layout and the same content.

Reviewer 2 ·

Basic reporting

The paper is very well written, organized and clear to read. Text is concise and cohesive. Moreover, authors did a good job in keeping the literature review closer to the scope of the paper. Some minor feedback on how to improve the tables, and include more visual presentation of the quantitative are discussed in the detailed review below.

Experimental design

There are two main problems with the design, further detailed below in the Comments to author section. Both problems are connected to ethical guidelines and data collection practices. Problem 1: Authors are not clear on the ethical guidelines that they follow. The text is not clear, whether the participants were given the option to opt-out. Even if the experiment is not graded, it seems that they "needed to participate" in order to pass a course. Even if they needed to participate, participants should be given the option to participate in the experiment (i.e., to complete the course), but ask that their data is not used for research or is not published. Problem 2: Collection of personal data. Authors collect and publish very personal data from the participants. Particularly for age and gender this would risk the anonymization of the data, given that some combinations of those are unique. Going through the paper, gender and age have no weigh in the conclusions. The data is used to claim consistency between both groups of participants, but authors can use the other data collected to argue for that (other rows of Table 6). There are also some concerns with the construct used for gender (0 for females, 1 for males), which violates recent guidelines to collect gender data (established from the field of gender studies, medicine and psychology, detailed in comments below).

Below, I present the detailed parts of the paper where I found both problems, along with a few suggestions for authors to consider.

Validity of the findings

The statistical analysis performed has several issues. Authors neglected to check assumptions of the test, and chose unsuitable test and correlation scores for their analysis. These checks have been widely discussed in the Empirical Software Engineering field. This is reflected on a lack of an analysis on conclusion validity threats. The authors did a great job with their data package, that allowed me to rerun all tests and even run with the right tests. Fortunately, the conclusions and findings remain overall the same, but the text should reflect and motivate these choices better. Moreover, the analysis with metrics (Section 5) is also subjective to validity threats (already mentioned by authors in the text) that should be further discussed.

Below I provide suggestions on how to fix both the statistical analysis and the metrics discussion.

Additional comments

Summary of the paper:
The paper evaluates the effect of four service-based patterns in an experiment with 62 students. The patterns are evaluated in terms of effectiveness and efficiency related to modifiability of the system. The analysis reveal that the use of patterns does not show a significant difference in modifiability (better or worse), except for a specific difference on one of the tasks performed (Task 3). Authors complement this evaluation doing an analysis on internal quality metrics of the system.

Strengths:
* Great effort in controlling the main threats when designing and running the experiment.
* Providing all data available enabled a thorough validation of the findings and the paper. This is certainly commendable and hopefully referenced by future empirical studies.
* Good contribution on the lessons learned (Section 8) to foster reproduction of the experiment and encourage more quantitative studies in the field.

Weaknesses:
* There are very pertinent open questions about the ethical protocols used by authors that require clarifications.
* Collection of personal data from participants is not properly motivated or justified.
* The statistical analysis were based on the wrong tests, introducing conclusion validity threats.
* The reporting and packaging of the experiment can be improved.

Summary of the review:
First and foremost, authors should be commended on the effort in making all the data available. Certainly, it allowed this review to go in a much deeper level, to identify and validate the findings (which is how science should be done). There are, unfortunately, major problems with the paper, but identifying and acting on these problems is not an issue, since the paper and the data (i.e., the research itself) allows that. Sharing the data allowed me to rerun the analysis and do "my own" check of data and conclusions, and I can say that, even though, the statistical analysis performed by authors is wrong, I checked that the conclusions would have been the same had the right statistical toolkit had been applied. In summary, I highlight three main problems with the paper: i) Ethical guidelines, ii) Anonymity of data, iii) Statistical analysis and iv) Metrics evaluation. Problems 'ii' and 'iii' are an easy fix, and I provided a variety of suggestions below; problem 'iv' is not that big, but authors should be aware of it and address it; problem 'i' is a bit more tricky and hopefully authors can state which ethical guidelines they followed, and clarify some aspects of the participant's consent. Since the paper is moving forward and encourages reproducible quantitative evaluation on the topic, the contributions are relevant to our field. Therefore, I suggest a "Major revision" so that authors can fix all 4 issues, even though the effort in fixing those are straightforward.

Problem 1: Ethical guidelines
Regarding student's particiation, authors state (Page 7/20): "Bachelor students that needed to participate in an experiment ... while the experiment participation was not graded, they nonetheless had motivation to participate to pass the course." To me that statement is either conflicting or confusing. Did students *need* to participate in an experiment to pass the course? Perhaps I misunderstood, but does it mean that students *had to* participate, but were not graded based on their performance? Could they still pass the course *without* participating in the experiment? If the answer is 'yes', please clarify so. If the answer is 'no', refer to the comment below:
According to ethical guidelines [Badampudi 2017; Singer and Vision, 2002], one must ask participants for consent while *minimizing the possibility of coercion or undue influence*. Did participants sign any form? Students should be given the option to participate in the experiment (i.e., pass the course) but also ask researchers to *not* use their data (i.e. opt-out of the study). Was that done? If 'yes', please clarify so in the text. If 'no', then the experiment has severe ethical implications since students were coerced to participate and authors should consult with an ethical committee on how to proceed.
Specific parts supporting the arguments above are:
Singer and Vision, 2002; check Sections II, points 1 and 5;
Badampudi 2017; check pages: 1172-1773

Problem 2: Anonymity of collected data.
Authors do not motivate the purpose of collecting the participant's personal information (age, gender, etc.). There is no relevance in disclosing such information, and no need for it in this study, since it risks breaking anonymity and confidentiality of the data collected [Badampudi 2017; Singer and Vision, 2002].

Particularly, I am concerned when authors ask for gender; the survey file does not suggest any value for gender (which is good, in my opinion) so I am assuming authors did not provide "options". On the other hand, the R script calculates the % of male participants as a mean of "1" values. That construct is wrong in many levels. First, the script assumes there are only "0" (females) and "1" (males). That in itself is a bad construct, since that formula conveys the idea that "only males count". More important, two values does not capture the diversity of gender (cis-, trans-, etc.). There are numerous discussion on how to collect data on gender (identity) or (biological) sex (note that both are very different for survey purposes - see https://news.stanford.edu/2015/08/19/gender-sex-surveys-081915/). Particularly, a good summary of the discussion can be seen here (highly recommended read!): https://www.surveygizmo.com/resources/blog/how-to-write-survey-gender-questions/ or in this source: https://hrc.org/resources/collecting-transgender-inclusive-gender-data-in-workplace-and-other-surveys

Academic discussions on the topic are very much on the humanities field and a current topic in LGBTQ+ studies such as Reisner et al., available here: https://www.ncbi.nlm.nih.gov/pubmed/25030120. Moreover, I am assuming that students were given the option to not disclose their personal information (i.e., they could leave it blank). I have access simply to the questions, but not the web-form itself (or I could not find it in the provided packages); however, all students provided their personal information (all fields are filled with 0 or 1). Which is not a good indication since, again, it seems that they were *not* given the option to opt-out.

Finally, gender information is mostly relevant for studies focused on psychology or cultural aspects, whereas biological sex is an object of study for medical studies. Since this paper does not seem to fall on neither categories, I strongly suggest that authors remove the personal details (at least gender and age) of the students from the paper, the final dataset and from the tables altogether, since this is sensitive data that can be used to break anonymity of the data (unlikely, but not entirely impossible).

For instance, if the majority were male (Table 3), one can simply search for a 30 year-old female student at the 2 semester of the BSc. on Software Engineering. Even if many are available, that still narrows it down; or the only 19 year-old female who is from the "Other" program that participated in this experiment.

Problem 3: Statistical analysis.
In short, all choice of tests were not suitable for the data collected.

3.1. Correlation analysis:
Authors use Spearman's rho to analyze correlation, even though it is not appropriate for this analysis. So, Spearman uses the rankings of the observation. Albeit Spearman's rho is non-parametric, there are two main assumption, one of them being a monotonic relationship between the variables. If one does a scatterplot of the effectiveness over any of the other variables, one can see that the relationship is not monotonic.
More troubling for the effectiveness, is that Spearman's rho does not respond well to ties. For instance, authors state that there is a correlation between effectiveness and attendance to the intro lecture, but since there are only two categories on the "ordinal data" for the intro column, there will basically be two ranks, which R should calculate as the average rank of all repeated values (e.g., execute: rank(data$intro)). Nonetheless, there are basically only 4 possible values for effectiveness, which then again, will lead to many tied ranks in the data. I checked R's documentation for the corr.test method used, and it does seem they correct for ties (but only the source code can really tell).
I reran the tests using kendall's tau (a better choice since its less senstivitive to these discrepancies in ranks), and the results are similar for effectiveness vs. intro. So, the spearman implementation should be correct in R. However, the choice does not cater for the monotonic relationship.
In short, I suggest that authors revise the choice of correlation score. I suggest Kendall's tau, since it can lead to slightly similar inferences as Spearman, and is more permissive with assumptions. More importantly, I am not sure the choice of constructs (particularly effectiveness), is suitable for using these correlations (there aren't that many values that effectiveness can take), to an extent where I argue that the correlation found cannot lead to any discussion around causation. So, consider *also* emphasizing that in the validity threat section.
3.2 Hypothesis testing
Regarding the hypotheses testing, there are two severe problems that invalidates the statistical results of the test:

Problem 1: Authors used a Welch t-test for the hypothesis testing. That is the wrong choice for their data. Welch's t-test assumes that the samples are normally distributed which is not the case for effectiveness or efficiency. Doing a quick Shapiro-wilk test (or Anderson Darling) for normality, quicly reveals that the data are not normally distributted. Moreover, plotting a simple histogram shows that the distribution is, in fact, quite uniform for effectiveness. The visual distribution of the efficiency is slightly better, but it still fails the Shapiro-Wilk test.

Therefore, any parametric test or anything assuming normality can be quite risky to use. I suggest that authors revise the text. They should: i) properly motivate the choice of test making sure that the assumptions are met, hence introducing information on analysis of normality; ii) turn to non-parametric choices, that seems a better choice for their data. Perhaps a simple Mann-Whitney test could be enough for them, even though there will again be many ties in the ranks. I did a quick check, and for a two-sided test, the mann-whitney test indicates no differences for effectiveness (p-value = 0.09, so nothing changes), but with 0.1 alpha, there is a slightly significant difference between duration.

Problem 2: I checked the scripts, and the actual calculation *does not match the stated hypotheses*. The t-test method, in R, uses by default a two-sided test, often expressed as a "H1: a != b". Authors use a one-sided test, meaning that only half of the p-value should be accounted for the given level of confidence. For instance, for effectiveness, the p-value reported is: 0.915, when in fact, the "right" p-value should be: 0.54 (or 0.4574 if I misunderstood the script). This should be done by defining either "greater" or "less" on the alternative field in the t.test R function. So, on one side, the conclusions should not change since, still, there are no differences (high p-value); on the other side, authors don't need to bother, since the choice of Welch t-test was wrong in the first place. I suggest authors review the alternative hypotheses, and make sure the right calculations are being used by the R functions when choosing another test.

Checking assumptions behind parametric tests has been already discussed by [Arcuri and Briand, 2011], other risks raised by [Kitchenham et al., 2019] and the risks of using null hypothesis significance testing altogether was raised by [Sheppard, 2018] and [de Oliveira Neto et al., 2019]. I recommend authors the recent analysis done by [de Oliveira Neto et al., 2019] where its authors discuss the main pitfalls with statistical analysis in SE, but provide guidelines on how to avoid these pitfalls, along with a suggestion of tests that can be used.

Problem 4:
This is not as challenging as the other problems. On section 5, authors evaluate the different versions in terms of different internal quality metrics. This is tangential to the analysis with students, so it is appreciated to introduce a new perspective on the analysis. However, if I understood correctly, authors evaluate internal quality of artefacts created by themselves (since my understsanding is that the implementation of the services was done by researchers?!). Authors do acknowledge this issue under their external validity threat when they say: "experiment tasks are still [...] subjective bias".

Since the artefacts were developed to better fit the experiment context (as should be the case, since many aspects should be controlled in an experiment with people), the metric data, however interesting is not reliable or representative. I suggest that authors to i) re-run this analysis on perhaps an existing project (or more) that has such patterns. Even though I understand the difficulties of finding artefacts, *an alternative* should be to ii) have non-researchers or more senior students implement a few of these patterns and then evaluate them instead. In either case, this, unfortunately, means some revision of Section 5.

As a compromise, authors could try to keep section 5, but reduce the strength of their claims throughout the discussion, and explicitly highlight these limitations on this analysis (i.e. not representative) as a construction and external threat (i.e., expand a bit the current external validity text). The future work text can also build on this analysis to implement the suggestions 'i' or 'ii' above.

Minor comments:
M1 - Authors define that the purpose of their experiment is "modifiability" (P.6, L.68), but that is still a bit too open and not really a purpose (rather it's just the quality characteristic targeted). To further clarify the research objective, I suggest authors include the actual purpose in their description, such as one of the following: "for the purpose of characterizing/comparing/evaluating/predicting/controling modifiability" (as suggested by Wohlin et al., 2012 - Table 7.3).

M2 - Regarding the available system used in the experiment, I had trouble executing the "_scripts" folder at first on a MacOS. The README did not provid einstructions on how to make them work on a MacOS. My MacOS did not have the gnome-terminal, so I basically just commented out the first and last line. Also, the README suggests execution of the *srv-run.sh files, but the system needs to be built at least once so the target files can be found. I could not execute the WebUI also due to problems with npm.

I suggest that authors edit their README file, to provide pointers on how ot execute either on Windows or MacOS. If it works on MacOS, should work on Linux as well. Nonetheless, the scripts and analysis data were executable and editable, so regardless, the attitude and organization of the authors is commendable for fostering reproducibility.

M3 - Authors state on Page 4/20: "And second, if the patterns’ advantages materialize even with inexperienced developers that have little or no conscious knowledge of them, *their effect on evolvability must be really significant*.". That seems like a fair assumption, but it is misleading due to the use of "really significant". Certainly, would be more beneficial, but it is conjecture to assume the magnitude of this significance. I suggest authors rephrase the sentence to avoid overstating the magnitude of the effect.

M4 - Authors use the term (Page 7/20, line 204): "dependent response variables". The terminology is either "dependent" or "response" variable. Choose one.

M5 - Simply an aesthetic suggestion (feel free to ignore this one): reading through the formal hypotheses is not easy. Authors could format the text in a table (grid) with two columns (null and alternative hypothesis). Each row would be a pair of null/alternative hypotheses. In each cell (or different cells in a multi-row table in latex), authors could include the text + equation for the corresponding hypothesis. This would make it easier to visualize how there are 1 pair for effectiveness (also that this is a one-tailed test), and the distinction with efficiency per task (H1-3) and all tasks (H4)

M6 - When presenting the mean values in tables 3 and 4, authors do not clarify whether the 1-10 scale means. Does 10 mean very experienced on the topic? I suggest that authors clarify the scale both in the questionnaire file (zenodo) and the text or table caption.

M7 - On Table 4, the Mean Duration for all Tasks is ambiguous between i) the mean duration of each task (i.e., the mean is calculated over all task entries in the data), and ii) the mean duration for people who complete all tasks (i.e., how long on average it takes to do all 3 tasks).

I understand that authors and the script report on 'i', so I suggest authors clarify in the table (caption or footnote or row name) to remove the ambiguity. Our of curiosity, I calculated 'ii' in the scripts and the values were: version 1: 4530s; version 2: 3819s; (ca 12 minutes difference). I suggest authors to include this information. Certainly, the calculations only considered rows where effectiveness was 1 (i.e., all 3 tasks completed).

M8 - Authors state: "Roughly 30% were not able to solve any task.". The percentage should be complemented by the absolute numbers per version. For instance, I checked the data and it shows that the number is 10 for v1 and 11 for v2. I suggest that author add that information to the text, to show that there was no specific group that could not completely (it was virutally the same)

M9 - Authors state: "The only notable exception for this was task #3 which was perceived as about 40% less difficult by the pattern group #2." This is insightful, but I believe the statement is incomplete. Since only 25% of participants finished task 3 (10 for v2, and 7 for v1), perhaps those 40% are indeed not representative?! I personally cannot tell, I would have to analyse the data to assess what is this 40% difference in absolute terms (i.e., is it 4 points on the 1-10 scale easier?). I looked at the histograms of the ex3.diff of each version and that was more insightful. I suggest that authors elaborate on this difference providing more absolute terms to complement the current percentage value (40%). Perhaps introducing a section with some histograms of specific distributions would be more insightful than simply reading the numbers.

M10 - Page 10/20, lines 273,274 authors report on a multiple test for each task using Bonferroni correction. Also, authors report on a power analysis. I could not find thoses tests on the provided scripts or package. I suggest that authors add those tests to the script. It would be great to add, in the script as a comment, the corresponding page where or section where the test is being reported in the text.

M11 - As an aesthetic, but very relevant suggestion, Table 6 should be re-designed. The choice of colors is not good for many reasons. First and foremost, red and green is not a good combination for colorblind readers, and secondly, the distinction lies, currently on the hue, which is undistinguishable when printing on greyscale. I suggest that authors change to a colorblind friendly pallete that also have different luminosity/shading so that black-white prints can also highlight the differenes.

References:
F. G. de Oliveira Neto, R. Torkar, R. Feldt, L. Gren, C. A. Furia, Z. Huang. Evolution of statistical analysis in empirical software engineering research: Current state and steps forward. 2019. Available on arXiv: https://arxiv.org/abs/1706.00933

D. Badampudi, "Reporting Ethics Considerations in Software Engineering Publications," 2017 ACM/IEEE International Symposium on Empirical Software Engineering and Measurement (ESEM), Toronto, ON, 2017, pp. 205-210. doi: 10.1109/ESEM.2017.32

J. Singer and N. G. Vinson, "Ethical issues in empirical studies of software engineering," in IEEE Transactions on Software Engineering, vol. 28, no. 12, pp. 1171-1180, Dec. 2002. doi: 10.1109/TSE.2002.1158289

S. L. Reisner, K. Biello, J. G. Rosenberger, S. B. Austin, S. Haneuse, A. Perez-Brumer, D. S. Novak, M. J. Mimiaga. Using a two-step method to measure transgender identity in Latin America/the Caribbean, Portugal, and Spain. Arch Sex Behav. 2014 Nov;43(8):1503-14. doi: 10.1007/s10508-014-0314-2. Epub 2014 Jul 17.

M. Shepperd. Replication studies considered harmful. 2018. Available in: https://arxiv.org/abs/1802.04580

B. Kitchenham, L. Madeyski, and P. Brereton. 2019. Problems with Statistical Practice in Human-Centric Software Engineering Experiments. In Proceedings of the Evaluation and Assessment on Software Engineering (EASE '19). ACM, New York, NY, USA, 134-143. DOI: https://doi.org/10.1145/3319008.3319009

---

## Round 0.2 · accepted · Accept

Both reviewers are now (almost) completely happy with the state of the manuscript. Please do address the small comment related to the bibliography during copy editing.

Reviewer 1 ·

Basic reporting

As written in my review for the original submission of this paper, I am very positive regarding the actual content of the paper.

Experimental design

I had only one major comment in my original review, asking for the discussion of standard deviations wherever appropriate. This has now been added.

As far as I can judge, the comments by the second reviewer have also been handled in a satisfying way. It should be noted that it seems as if the second reviewer has way more knowledge about this kind of studies than I do.

Validity of the findings

My major comment regarding the validity of the findings addressed that the developers participating in the study were quite junior. The according discussion (which was already fine before) has now been further extended.

Additional comments

Actually, I recommend not to use bold font to highlight single terms, since this decreases readability. Instead, italics should be used.

The authors should check again their references, since the level of detail given in the single entries is very heterogeneous. Just some examples - as written in my review to the original submission, the authors need to check all references carefully:
* Almost all articles are missing page numbers, a lot of journal articles are missing volume and issue information
* Uppercase and lowercase lettering of the titles does not seem to follow any pattern
* Some conference names include a "Proceedings of", but most don't. Some conference names are only given as abbreviations (e.g., Rud et al. 2006), while most are given completely.
* "Communications in Computer and Information Science" is given twice for the paper by Hegedüs et al., while the actual booktitle is missing.
* Missing blank in Kassab et al. (2012)
* Publisher names are given in an inconsistent way
* etc. etc.

Reviewer 2 ·

Basic reporting

The new version is improved in terms of presentation with clearer tables and better description of formal hypotheses, data and statistical results.

Experimental design

The main hindrances have been addressed and carefully explained by authors. The design and the analysis now are more suitable to the data collected and hypotheses investigated.

Validity of the findings

Authors were very transparent with the main threats to validity. Also, the sharing of the experiment's data, scripts and artefacts significantly add to the reliability of the experiment since it allows other researchers to re-run the analysis done by authors.

Additional comments

Authors did a great job both in addressing all concerns I had regarding the paper and in providing detailed information regarding their changes and updates. Particularly, I appreciate that authors were thorough when describing the ethical concerns regarding their data collection in the new version of the paper. The new version of the analysis is also more suitable to the data collected, as well as the updates on the presentation. In summary, the paper was improved and addressed the main shortcomings, so I recommend an Accept to this paper without the need of further updates.